# A Variational Synthesis of Evolutionary and Developmental Dynamics

**DOI:** 10.3390/e25070964

**Published:** 2023-06-21

**Authors:** Karl Friston, Daniel A. Friedman, Axel Constant, V. Bleu Knight, Chris Fields, Thomas Parr, John O. Campbell

**Affiliations:** 1Wellcome Centre for Human Neuroimaging, Institute of Neurology, University College London, London WC1E 6AP, UK; k.friston@ucl.ac.uk (K.F.);; 2Department of Entomology and Nematology, University of California, Davis, Davis, CA 95616, USA; 3Active Inference Institute, Davis, CA 95616, USA; blanket@activeinference.institute; 4Theory and Method in Biosciences, The University of Sydney, Sydney, NSW 2006, Australia; axel.constant.pruvost@gmail.com; 5Department of Biology, New Mexico State University, Las Cruces, NM 88003, USA; 6Allen Discovery Center at Tufts University, Medford, MA 02155, USA; fieldsres@gmail.com; 7Independent Researcher, Victoria, BC V8V 1X4, Canada

**Keywords:** self-organisation, nonequilibrium, variational inference, Bayesian, particular partition, evolution, natural selection, Markov blanket, renormalisation group

## Abstract

This paper introduces a variational formulation of natural selection, paying special attention to the nature of ‘things’ and the way that different ‘kinds’ of ‘things’ are individuated from—and influence—each other. We use the Bayesian mechanics of particular partitions to understand how slow phylogenetic processes constrain—and are constrained by—fast, phenotypic processes. The main result is a formulation of adaptive fitness as a path integral of phenotypic fitness. Paths of least action, at the phenotypic and phylogenetic scales, can then be read as inference and learning processes, respectively. In this view, a phenotype actively infers the state of its econiche under a generative model, whose parameters are learned via natural (Bayesian model) selection. The ensuing variational synthesis features some unexpected aspects. Perhaps the most notable is that it is not possible to describe or model a population of conspecifics per se. Rather, it is necessary to consider populations of distinct natural kinds that influence each other. This paper is limited to a description of the mathematical apparatus and accompanying ideas. Subsequent work will use these methods for simulations and numerical analyses—and identify points of contact with related mathematical formulations of evolution.

Dedicated to the Memory of John O. Campbell.

## 1. Introduction

This paper is an attempt to show that some fundaments of theoretical evolution—and (neuro)biology—emerge when applying the free energy principle to dynamical systems with separation of temporal scales. It offers a technical and generic treatment with minimal assumptions or commitments to specific biological processes. As such, it does not borrow from established constructs in evolutionary theory; rather, it tries to show how some of these constructs are emergent properties, when seen through the lens of the free energy principle. In subsequent work, we will use the ensuing variational synthesis to consider established—and current—evolutionary theories. Our aim in this paper is to introduce a formalism that may be useful for addressing specific questions—about evolutionary or developmental dynamics—using analytic or numerical recipes that have proven useful when applying the free energy principle in other fields.

A key phylogenetic process—underlying the development and diversification of species in evolutionary time—is known as natural selection, regarded by some as the central organizing principle of biology. While Darwin conceived of natural selection in terms of heredity, variation, and selection [1,2], he only detailed selection, as the mechanisms of heredity and variation would not be understood for some time [3,4]. The integration of Mendelian genetics with natural selection in the early twentieth century was followed by an integration with molecular genetics [5] in the mid-century to form Neo-Darwinism, or the modern synthesis. The modern synthesis, along with the selfish gene hypothesis—put forth in the 1970s [6]—provide a largely gene-centric view of Darwinian evolution that dominates the current perspective.

This gene-centric view of evolutionary biology has remained largely disconnected from phenotypic processes that impact organisms in developmental time [7,8]. Lewontin characterised this disconnect—between genetic and phenotypic understanding—as the major challenge facing the field [9]. While some progress has been made in the following fifty years, biologists continue to highlight the gaps remaining for modelling biology as a single integrated process over multiple scales [10,11,12,13]. By ‘gene-centric’, we refer not just to theories of sequence evolution [14], but also to the central role genes (or summary statistics of genes) play either explicitly or implicitly in accounts of phenotypic evolution. For instance, the Price Equation [15] and the closely related replicator equation [16] of evolutionary game theory express the relationship between the changes in (the average of) some phenotypic trait over time. This gene-centric view relies upon a mapping between that trait and the genetic material passed from generation to generation but focuses upon the phenotypic effects of genes as opposed to the alleles themselves. Similarly, adaptive dynamic approaches [17] typically focus upon ecological interactions at a phenotypic level. The modern focus upon phenotypic traits reflects the importance of the interaction between a phenotype and its environment in determining fitness. However, it is important to note that such perspectives do not conflict with the central role of genetic inheritance, and implicitly score the fitness of genotypes in terms of the phenotypes they imply.

An organism inherits a set of instructions for growth and development (i.e., an extended genotype) that is, in essence, a prediction about the niche environment (including temperature, humidity, chemical composition, available resources, statistical patterns, etc.). Interrogating the phrase ‘survival of the fittest’ leads to the understanding of ‘fittest’ as organisms that are the best ‘fit’ to their niche environment [18]. For example, a bacterium from thermal hot springs will fail to thrive in a cool pond because its genotype does not accurately predict the niche environment. Therefore, ‘fitness’ is relative to the niche, where slow phylogenetic processes have selected for an extended genotype that enhances the growth and proliferation of organisms in the environment where the corresponding species expects to find itself.

An organism can also ‘fit’ itself to the niche through adaptation (i.e., action, learning, and development) during its lifetime. For example, a bacterium that normally subsists on sulphur reduction—but can also survive through reducing oxygen—will outlast its sulphur-dependent competitors in an environment that is devoid of sulphur. Such an organism can adapt to its environment through learning and optimising for oxygen reduction, thereby increasing its fit to the niche and, implicitly, its capacity to reproduce in a high-oxygen environment. In this way, the phenotypic processes can enhance the fit of organisms to their environment in developmental time, and through reproduction, phenotypic processes can lead to the enhancement of fit in evolutionary time (i.e., across generations). As the (extended) genotype of organisms produces phenotypes, phylogenetic processes over evolutionary time also impact phenotypic (ontogenetic) processes in developmental time.

Here, we offer a synthesis of evolution and development through a mathematical framework that unifies slow, multi-generational (phylogenetic) processes with single-lifetime, phenotypic (developmental and behavioural) processes using the same principles, as they apply to each temporal scale. The ensuing variational account of evolution focuses on the coupling between phylogenetic processes at evolutionary timescales and ontogenetic processes over phenotypic lifetimes. In principle, this abstract treatment is agnostic to specific mechanisms, and could be applied to biological as well as non-biological systems provided their ‘fitness’ depends upon events during a lifetime, and where this fitness influences dynamics over a generational scale. This multiscale account foregrounds the circular causality that arises from the implicit separation of timescales [19].

In brief, we consider slow phylogenetic processes (*natural selection*) as furnishing top-down constraints (i.e., top-down causation) on fast phenotypic processes (*action selection*). In turn, the active exchange of the phenotype with its environment provides evidence that is assimilated by natural selection (i.e., bottom-up causation). This ontological account is licensed by describing both phylogenetic and phenotypic processes as selecting (extended) genotypes and (extended) phenotypes [7,20] with the greatest fitness, where fitness is quantified with (free energy) functionals of probability density functions (a functional is a function of a function).

This formulation means that natural selection and action selection can be described as updating probabilistic beliefs at phylogenetic and phenotypic scales, respectively: namely, learning and inference [21,22,23]. This separation of scales affords an interpretation of *natural selection as Bayesian model selection* [24,25,26], while *action selection becomes planning as inference* [27,28,29,30]—both (appearing to) optimise the same fitness functional: namely, Bayesian model evidence or marginal likelihood. A narrative version of this account can be told from the point of view of the genotype (from the bottom up) or the phenotype (from the top down):

**From the perspective of the genotype**, we can consider evolution as belief-updating over generations, where the *belief* in question corresponds to a probability density over extended genotypes (henceforth, genotype). This belief-based model of allelic change is analogous to treatments of evolution in terms of changes in allele frequencies from generation to generation [15]. This belief updating can be described by the probability of a genotype appearing in subsequent generations, in a way that depends lawfully on the marginal likelihood of extended phenotypes (henceforth, phenotype) in the current generation. *The basic idea is that the genotype parameterises or encodes a generative model, which the phenotype uses to infer and act on its environment.* On this view, evolution can be regarded as testing hypotheses—in the form of generative models—that this kind of phenotype can persist in this environment. These hypotheses are tested by exposing the phenotype to the environment and are rejected if the phenotype ‘strays from the path’ of a persistent phenotype. In this way, the evolutionary process selects models or hypotheses about persistent phenotypes for which it has the greatest evidence. In short, natural selection is just Bayesian model selection [25,26,31,32].

**From the perspective of a phenotype**, each conspecific is equipped with a generative model and initial conditions that underwrite its epigenetic, developmental and ethological trajectories. The states of the phenotype trace out a path through state-space over its lifetime. These phenotypic states encode or parameterise beliefs about environmental states—and the way the phenotype acts. This parameterization leads to active inference and learning, in which the phenotype tries to make sense of its world and—through a process of belief updating—to realise the kind of creature it thinks it is. (We use the term ‘thinks’ in a liberal sense here and do not mean to imply that all living entities have explicit existential thoughts.) More precisely, what we mean is that these entities behave as if they hold a set of beliefs about the sort of entity they are (e.g., the meta-Bayesian stance as considered in [33]). In virtue of its genetic endowment, it thinks it is a persistent phenotype. If endowed with a good generative model of its environment [34], it will persist and supply evidence of its ‘fit’ to the environment (i.e., ‘fitness’); namely, evidence (i.e., marginal likelihood) that has been accumulated by the slow evolutionary process.

What follows is a formal version of this narrative that calls upon some standard results from statistical physics. The resulting synthesis is both dense and delicate, because it tries to account for coupling between a phenotype and its econiche—and the coupling between phenotypic and phylogenetic processes—using the same principles. Specifically, we describe a variational synthesis that calls on the path integral formulation of stochastic dynamics, the apparatus of the renormalisation group, and the Poincaré recurrence theorem. The ensuing synthesis considers natural selection and action selection as emergent properties of two random dynamical processes unfolding at slow (phylogenetic) and fast (phenotypic) timescales. The key aspect of this synthesis is that both processes have an attracting set (a.k.a., pullback attractor) or steady-state solution [35]. These solutions correspond to an *evolutionary stable state* [36] and a *nonequilibrium steady-state density* [37] over phylogenetic and phenotypic states, respectively. By describing these steady states in terms of a phylogenetically encoded generative model—namely, a joint density over the paths of the phenotype and its environment—one can recover an ontological description of how the two processes inform, and are informed by, each other.

Some of the analysis presented in this paper follows that in [21,22,23], which also appeals to the notion of a renormalisation group. These treatments are based upon the emergence of separable timescales and the interpretation of the dynamics at each scale in analogy with inference and learning processes. The key differences are as follows. The renormalisation in [21] depends upon a reduction in the number of degrees of freedom with learning, whereas our formulation depends upon a partitioning operation as part of the renormalisation. The difference in timescales between variables in [21] emerges from the structure of the neural network used, whereas it is a direct consequence of the reduction operator implicit in our choice of renormalisation. Finally, we extend our analysis to sentient phenotypes, whose dynamics can be interpreted explicitly in terms of Bayesian belief-updating. We conclude with a numerical study, illustrating the basic ideas with synaptic selection in the brain.

## 2. A Variational Formulation

We assume that evolution can be described with two random dynamical systems, describing phylogenetic (evolutionary) and phenotypic (particular) processes, respectively. The idea is to couple these systems using the apparatus of the renormalisation group [38,39,40] to map from fast phenotypic dynamics to slow phylogenetic dynamics in evolutionary time.

This mapping rests upon a dimension reduction and coarse graining or grouping operator (**RG** for Renormalisation Group) that maps the path of a phenotype $\tilde{\pi }$ to relevant variables at the evolutionary scale $\bar{\pi }=R\circ \tilde{\pi }$. On this view, bottom-up causation is simply the application of a reduction operator, $R\circ \tilde{\pi }$, to select variables that change very slowly. Top-down causation entails a specification of fast phenotypic trajectories in terms of slow genotypic variations, which are grouped into populations, $G\circ \bar{\pi }$, according to the influences they exert on each other. The implicit separation into fast and slow variables can be read as an adiabatic approximation [41] or—in the sense of synergetics—into fast (dynamically stable) and slow (dynamically unstable) modes, respectively [42]. This separation can also be seen in terms of vectorial geometric formulations [43]. Please see [21], who deal carefully with the separation of time scales by analogy with temporal dilation in physics. Intuitively, this analogy rests upon the idea that time can be rescaled, depending upon whether we take the perspective of things that move quickly or slowly.

The final move is to express the dynamics—at fast and slow levels—in terms of functionals that have the same form. These functionals are functions of probability densities that can be read as Bayesian beliefs. Expressing the dynamics in this way allows one to interpret phenotypic dynamics as active inference and learning, under a generative model that depends on the extended genotype. In other words, one can interpret the phylogenetic state as inferring states of the environment over evolutionary time. Crucially, the extended genotype accumulates evidence for its phenotype, thereby evincing a form of Bayesian model selection or structure learning [25,44,45,46,47,48]. For an analogous thermodynamic treatment, please see [22], who refine and extend the free energy formulation of [49]. In the context of learning dynamics, a thermodynamic free energy was derived in [50]—using the maximum entropy principle [51,52]—and later applied to study phenomenological models of evolution [22]. Please see [50,53,54] for further discussion in terms of neural networks and information theory.

### 2.1. Particular Partitions

There are many moving parts in this formulation because it tries to account for the behaviour of ‘things’ [55] and how this behaviour underwrites the emergence of ‘kinds’ (e.g., individuals and populations) at nested (i.e., developmental and evolutionary) timescales.

We will use $[x(t)]\subset \tilde{x}$ to denote the history or path of a time-varying state. These paths are determined by state-dependent flow $f_{\bar{x}}(x)$, with parameters $\bar{x}\subset \tilde{x}$ that include initial states $x(0)=x_{0}\subset \bar{x}$. These parameters denote a (natural) kind.

Everything that follows rests upon a *particular partition* of states. A particular partition is considered necessary to talk about ‘things’, such as a ‘phenotype’ or ‘population’. In brief, a particular partition enables the (internal) states of some ‘thing’ to be separated from the (external) states of every ‘thing’ else by (sensory and active) blanket states [56,57,58,59,60]. In the absence of this partition, there would be no way of distinguishing a phenotype from its external milieu—or a population from the environment. In this setup, external states can only influence themselves and sensory states, while internal states can only influence themselves and active states. See Figure 1 for an influence diagram representing the coupling among internal, external, and blanket states:**States**: x=(η,s,a,μ). States comprise the *external*, *sensory*, *active* and *internal* states of a phenotype. Sensory and active states constitute *blanket* states b=(s,a), while *phenotypic* states comprise internal and blanket states, π=(b,μ)=(s,α). The *autonomous* states of a phenotype α=(a,μ) are not influenced by external states:**External states** respond to sensory and active states. These are the states of a phenotype’s external milieu: e.g., econiche, body, or extracellular space, depending upon the scale of analysis.**Sensory states** respond to fluctuations in external and active states: e.g., chemo-reception, proprioception, interception.**Active states** respond to sensory and internal states and mediate action on the environment, either directly or vicariously through sensory states: e.g., actin filaments, motor action, autonomic reflexes.**Internal states** respond to sensory and active states: e.g., transcription, intracellular concentrations, synaptic activity.

The evolution of these sparsely coupled states can be expressed as a Langevin or stochastic differential equation: namely, a high dimensional, nonlinear, state-dependent flow plus independent random (Wiener) fluctuations, ω, with a variance of 2Γ:(1)$$\dot{x}=f_{\bar{x}}(x)+\omega =\left[\begin{array}{l} \dot{\eta }\\ \dot{s}\\ \dot{a}\\ \dot{\mu } \end{array}\right]=\left[\begin{array}{l} f_{\bar{\eta }}(\eta ,s,a)\\ f_{\bar{s}}(\eta ,s,a)\\ f_{\bar{a}}(s,a,\mu )\\ f_{\bar{\mu }}(s,a,\mu ) \end{array}\right]+\left[\begin{array}{l} \omega_{\eta }\\ \omega_{s}\\ \omega_{a}\\ \omega_{\mu } \end{array}\right]$$
The flow per se can be expressed using the Helmholtz–Hodge decomposition [61] as follows:(2)$$\begin{array}{c} \;\;f_{\bar{x}}(x)=(Q-\Gamma )\nabla ℑ(x)\\ \left[\begin{array}{c} f_{\bar{\eta }}(\eta ,b)\\ f_{\bar{s}}(\eta ,b)\\ f_{\bar{a}}(b,\mu )\\ f_{\bar{\mu }}(b,\mu ) \end{array}\right]=\left[\begin{array}{cccc} Q_{\eta }-\Gamma_{\eta } & Q_{\eta s} & & \\ -Q_{\eta s}^{T} & Q_{s}-\Gamma_{s} & & \\ & & Q_{a}-\Gamma_{a} & Q_{a\mu }\\ & & -Q_{a\mu }^{T} & Q_{\mu }-\Gamma_{\mu } \end{array}\right]\left[\begin{array}{c} \nabla_{\eta }ℑ(\eta |b)\\ \nabla_{s}ℑ(b|\eta )\\ \nabla_{a}ℑ(b|\mu )\\ \nabla_{\mu }ℑ(\mu |b) \end{array}\right] \end{array}$$
Note that our appeal to an equation of this form means we have implicitly stipulated that there is a steady-state density or potential function that remains constant (or at least changes very slowly) over the timescale we are interested in. Equation (2) expresses the flow as a mixture of a dissipative, gradient flow and a conservative, solenoidal flow [62,63,64]. The gradient flow Γ∇ℑ depends upon the amplitude of random fluctuations, while the solenoidal flow Q∇ℑ circulates on the isocontours of the potential function called *self-information*, ℑ(x)=−lnp(x), where p(x) is called the *nonequilibrium steady-state density* or NESS density [37,65,66,67].

The particular partition above rests on sparse coupling between dynamic variables, c.f., [68,69], and evinces the notion of an ‘action-perception cycle’ between external and internal states [70]. The terms ‘external’ and ‘internal’ offer useful intuitions, but it is worth being cautious about overinterpreting these labels in spatial terms. For instance, it might seem that some ‘external’ variables such as ambient temperature might directly influence ‘internal’ variables such as the temperature within a cell. However, this intuition would not be an appropriate way of thinking about this system’s partition. Either we would have to assume that there is an intervening variable (e.g., the temperature within the cell membrane) or we would have to treat the internal temperature as a sensory variable, which itself influences internal variables such as the rates of enzymatic reactions. There is now an emerging literature asking about the appropriate ways to think of particular partitions in biology, including what is internal to a neuronal network [71], or a spinal reflex arc [72].

### 2.2. Ensemble Dynamics and Paths of Least Action

To describe dynamics at the phenotypic or phylogenetic scale, we first need to rehearse some standard results from statistical physics that furnish a probabilistic description of trajectories or paths at any scale. This description calls on the self-information of states x(t), generalised states $\vec{x}=(x,x^{\prime },\ldots )$, and paths, $\tilde{x}=[x(t)]$, where $D\vec{x}=(x^{\prime },x^{\prime \prime },\ldots )$ denotes generalised notion, and 2Γ is the covariance of generalised random fluctuations:(3)$$\begin{array}{l} ℑ(x)=-\ln p(x)\\ L(\vec{x})=-\ln p(\vec{x}|x_{0})=\frac{1}{2}[\ln|\Gamma |+(D\vec{x}-f(\vec{x}))\cdot \frac{1}{2\Gamma }(D\vec{x}-f(\vec{x}))+\nabla \cdot f]\\ A(\tilde{x})=-\ln p(\tilde{x}|x_{0})=\int dtL(\vec{x}) \end{array}$$

The first measure, ℑ(x), is the self-information or *surprisal* of a state, namely, the implausibility of a state being occupied. When the state is an allele frequency and evolves according to Wright–Fisher dynamics, this is sometimes referred to as an ‘adaptive landscape’ [73]. The second, $L(\vec{x})$, is the *Lagrangian*, which is the *surprisal* of a generalised state, namely, the instantaneous path associated with the motion from an initial state. In generalised coordinates of motion, the state, velocity, acceleration, etc., are treated as separate (generalised) states that are coupled through the flow [74,75]. Finally, the surprisal of a path $A(\tilde{x})$ is called *action*, namely, the path integral of the Lagrangian.

Generalised states afford a convenient way of expressing the path of least action in terms of the Lagrangian
(4)$$\nabla_{\vec{x}}L(\vec{x})+(\dot{\vec{x}}-D\vec{x})=0\Leftrightarrow \dot{\vec{x}}-D\vec{x}=-\nabla_{\vec{x}}L(\vec{x})\Leftrightarrow \dot{\vec{x}}(\tau )=D\vec{x}-\nabla_{\vec{x}}L(\vec{x})$$

The first equality resembles a Lagrange equation of the first kind that ensures the generalised motion of states is the state of generalised motion. Alternatively, it can be read as a gradient descent on the Lagrangian, in a moving frame of reference (second equality). When the Lagrangian is convex, solutions to this generalised gradient descent on the Lagrangian (third equality) necessarily converge to the path of least action. Denoting paths of least action with boldface:(5)$$\begin{array}{l} \dot{\vec{x}}=D\vec{x}=f(\vec{x})\\ \;\Leftrightarrow \nabla_{\vec{x}}L(\vec{x})=0\Leftrightarrow \vec{x}=\arg\min_{\vec{x}}L(\vec{x})\\ \;\Leftrightarrow \delta_{\tilde{x}}A(\tilde{x})=0\Leftrightarrow \tilde{x}=\arg\min_{\tilde{x}}A(\tilde{x}) \end{array}$$

Convergence is guaranteed by the quadratic form (i.e., convexity) of the Lagrangian, which inherits from Gaussian assumptions about random fluctuations. This gradient descent is sometimes described as convergence to the path of least action, in a frame of reference that moves with the state of generalised motion [76].

We can also express the conditional independencies implied by a particular partition using the Lagrangian of generalised states. Because there are no flows that depend on both internal and external states, external and internal paths are independent, when conditioned on blanket paths:(6)$$\frac{\partial^{2}f}{\partial \eta \partial \mu }=0\Rightarrow \frac{\partial^{2}L}{\partial \vec{\eta }\partial \vec{\mu }}=0\Leftrightarrow L(\vec{\eta },\vec{\mu }|\vec{s},\vec{a})=L(\vec{\eta }|\vec{s},\vec{a})+L(\vec{\mu }|\vec{s},\vec{a})\Leftrightarrow (\vec{\eta }⊥\vec{\mu })|\vec{s},\vec{a},x_{0}$$

In other words, blanket paths furnish a Markov blanket over internal paths. We will use this result later to disambiguate the role of active and sensory dynamics in sentient behaviour—i.e., active inference—of a phenotype. First, we have to establish a formalism for ensembles or populations of phenotypes. Here, we draw on the apparatus of the renormalisation group.

### 2.3. Different Kinds of Things

To deal with multiple ‘things’ (e.g., particles, phenotypes and populations), we first introduce a grouping operator **G** that partitions the states at the *i*-th scale of analysis into *N* particles on the basis of the sparse coupling implied by a particular partition. In other words, we group states into an ensemble of particles, where each particle has its own internal and blanket states. With a slight abuse of the set builder notation:(7)$$\left\{\pi_{1}^{\left(i\right)},\ldots ,\pi_{N}^{\left(i\right)}\}=G\circ \{x_{1}^{\left(i\right)},\ldots ,\underset{\pi_{n}^{\left(i\right)}}{\underset{︸}{\underset{s_{n}^{\left(i\right)}}{\underset{︸}{x_{j}^{\left(i\right)},\ldots ,x_{k}^{\left(i\right)}}},\underset{a_{n}^{\left(i\right)}}{\underset{︸}{x_{l}^{\left(i\right)},\ldots ,x_{m}^{\left(i\right)}}},\underset{\mu_{n}^{\left(i\right)}}{\underset{︸}{x_{o}^{\left(i\right)},\ldots ,x_{p}^{\left(i\right)}}}}},\ldots \right\}$$

The grouping operator means the external states of a given particle are the (blanket) states of remaining particles that influence it. See [55] for a worked example and numerical analysis. This grouping expresses the dynamics of each particle in terms of its sensory states—that depend upon the blanket states of other particles—and autonomous states—that only depend upon the states of the particle in question:(8)$$\begin{array}{c} \;\;\;{\dot{\pi }}_{n}=\left[\begin{array}{c} {\dot{s}}_{n}\\ {\dot{\alpha }}_{n} \end{array}\right]=\left[\begin{array}{c} f_{{\bar{s}}_{n}}(b_{n},\ldots ,b_{N})\\ f_{{\bar{\alpha }}_{n}}(b_{n},\mu_{n}) \end{array}\right]+\omega_{n}\\ \left[\begin{array}{c} f_{{\bar{s}}_{n}}\\ f_{{\bar{\alpha }}_{n}} \end{array}\right]=\left[\begin{array}{cc} Q_{s_{n}}-\Gamma_{s_{n}} & 0\\ 0 & Q_{\alpha_{n}}-\Gamma_{\alpha_{n}} \end{array}\right]\left[\begin{array}{c} \nabla_{s_{n}}ℑ(b_{1},\ldots ,b_{N})\\ \nabla_{\alpha_{n}}ℑ(\alpha_{n}|s_{n}) \end{array}\right] \end{array}$$

At this point, we pause to consider that the states in the particular ensemble have to be the states of some ‘thing’: namely, the states of a particle at a lower scale. This means that states must be the states of particles (e.g., phenotypic states) that constitute the particular states at the next scale (e.g., phylogenetic states). This recursive truism can be expressed in terms of grouping **G** operator—that creates particles—and a reduction **R** operator—that picks out certain particular states for the next scale:(9)$$\overset{R}{\to }\{{\tilde{x}}_{l}^{\left(i\right)}\}\overset{G}{\to }\{{\tilde{\pi }}_{n}^{\left(i\right)}\}\overset{R}{\to }\{{\tilde{x}}_{n}^{\left(i+1\right)}\}\overset{G}{\to }\{{\tilde{\pi }}_{m}^{\left(i+1\right)}\}\overset{R}{\to }$$

The composition of the two operators can be read as mapping from the states of particles at one scale to the next or, equivalently, from particular states at one scale to the next—in short, creating particles of particles, namely, populations. See Figure 2.
(10)$$\begin{array}{c} \overset{R\circ G}{\to }\{{\tilde{x}}_{l}^{\left(i\right)}\}\overset{R\circ G}{\to }\{{\tilde{x}}_{n}^{\left(i+1\right)}\}\overset{R\circ G}{\to }\\ \overset{G\circ R}{\to }\{{\tilde{\pi }}_{n}^{\left(i\right)}\}\overset{G\circ R}{\to }\{{\tilde{\pi }}_{m}^{\left(i+1\right)}\}\overset{G\circ R}{\to }\\ \left\{\pi (\tau {)}_{1}^{\left(i+1\right)},\ldots ,\pi {\left(\tau \right)}_{M}^{\left(i+1\right)}\}=G\circ \{R\circ {\tilde{\pi }}_{1}^{\left(i\right)},\ldots ,\underset{\pi_{m}^{\left(i+1\right)}}{\underset{︸}{\underset{s_{m}^{\left(i+1\right)}}{\underset{︸}{R\circ {\tilde{\pi }}_{j}^{\left(i\right)},\ldots ,R\circ {\tilde{\pi }}_{k}^{\left(i\right)}}},\underset{\alpha_{m}^{\left(i+1\right)}}{\underset{︸}{R\circ {\tilde{\pi }}_{l}^{\left(i\right)},\ldots ,R\circ {\tilde{\pi }}_{m}^{\left(i\right)}}}}},\ldots \right\}\\ {\bar{\pi }}_{n}^{\left(i\right)}=R\circ {\tilde{\pi }}_{n}^{\left(i\right)}\\ \left\{\pi_{m}^{\left(i+1\right)}\}=G\circ \{{\bar{\pi }}_{n}^{\left(i\right)}\right\} \end{array}$$

The reduction operator **R** typically selects relevant variables whose slow fluctuations contextualise dynamics at the scale below. Here, **R** simply recovers the states of a particle that are time invariant or that vary slowly with time (i.e., the initial states and flow parameters). This separation of timescales means that the lifetime of a particle (e.g., phenotype) unfolds during an instant from the perspective of the next scale (e.g., evolution). The separation of timescales could have been achieved without the grouping (partitioning) operator. We could simply have projected onto the eigenvectors of a dynamical system’s Jacobian, effectively taking linear (or nonlinear) mixtures of our system to arrive at fast and slow coordinates. However, all we would be left with are fast and slow continuous variables that have nothing of the character of the individuals, phenotypes, or populations in a system. In short, the grouping operator is key in identifying fast and slow ‘things’—as opposed to just fast and slow coordinates of a dynamical system.

In short, the renormalisation group operator creates particles of particles, retaining only particular variables that change very slowly and then grouping them according to their sparse coupling. This means that particles increase in their size from one scale to the next—in virtue of the grouping of particles at the lower scale—and change more slowly—in virtue of the coarse graining afforded by temporal reduction.

In an evolutionary setting, the existence of steady-state solutions—implicit in the Langevin formalism above—means that phenotypic dynamics possess a pullback attractor. This means their paths will return to the neighbourhood of previously occupied states. In other words, their ‘lifecycle’ will intersect with some Poincaré section in phenotypic state-space (possibly many times). We will take this intersection to be a mathematical image of persistence, which is underwritten by the flow parameters at any point in evolutionary time.

At the phylogenetic scale, we have a partition into populations of phenotypes based upon which phenotypes influence each other. At this slow scale, states can be read as characterising the ‘kind’ of ‘thing’ that has particular states at the scale below. We will, therefore, refer to states at this level as (natural) kinds, noting that the ‘kind of thing’ in question does not change at the fast scale. We can now rehearse the particular partition at the phylogenetic scale, noting that for a population to exist, it must have a particular partition. Here, a population corresponds to a set of particular kinds $x^{\left(i+1\right)}=(\eta ,s,a,\mu )$. These include *external*, *sensory*, *active*, and *internal* kinds.
**External kinds** of particles are phenotypes outside the population that change as a function of themselves and sensory and active kinds: c.f., the target of *niche construction*, from a molecular through to a cultural level, depending upon the scale of analysis [77,78].**Sensory kinds** mediate the effects of external kinds on the internal members of the population in question: e.g., nutrients or prey.**Active kinds** mediate the effects of internal kinds on external kinds: e.g., agents who mediate niche construction, from a molecular through to a cultural level, depending upon the scale of analysis.**Internal kinds** influence themselves and respond to changes in sensory and active kinds.

This concludes our formal setup. Next, we consider the coupling between fast phenotypic and slow phylogenetic dynamics. As in other applications of the free energy principle, this coupling emerges as a property of any phylogenetics that possesses an evolutionary steady state. In other words, the idea here is to identify the properties of a system that exists, as opposed to identifying the properties that underwrite existence. We will see that the emergent properties look very much like natural selection.

### 2.4. Natural Selection: A Variational Formulation

To specialise particular partitions to natural selection, we will associate autonomous (active and internal) kinds with the (extended) genotypes that constitute a population of *agents*, noting that there is no requirement for agents to belong to the same equivalence class—they just interact, in virtue of the sparse coupling that defines their grouping into a population. For example, some agents could be animals, and others could be plants.

At the phylogenetic scale, an agent is an autonomous kind from a particular population. At the phenotypic scale, the agent has particular (phenotypic) states, whose dynamics or paths depend upon its (genotypic) kind. For ease of notation, we will deal with a single population where the phenotypic state of the *n*-th agent, $\alpha_{n}^{\left(i+1\right)}$, will be denoted by $\pi_{l}^{\left(i\right)}$ (i.e., dropping the *m* in Figure 2). With this formalism in place, we can formulate the coupling between phenotypic and phylogenetic dynamics with the following lemma:

**Lemma** **1.***(Variational fitness): If, at non-equilibrium evolutionary steady state, the likelihood of an agent’s genotype* αn(i+1)=R∘π˜l(i) *is proportional to the likelihood of its phenotypic trajectory* π˜l(i) *(where\denotes exclusion),*

(11)$$\begin{array}{cc} {ℑ}_{n}^{\left(i+1\right)} & =ℑ(\alpha_{n}^{\left(i+1\right)}|\pi^{\left(i+1\right)}\backslash \alpha_{n}^{\left(i+1\right)})\\ & =A({\tilde{\pi }}_{l}^{\left(i\right)}|{\bar{x}}_{l}^{\left(i\right)}\subset \pi^{\left(i+1\right)})=A_{l}^{\left(i\right)} \end{array}$$
then the following holds:

An agent’s autonomous dynamics can be cast as a gradient descent on a Lagrangian, whose path integral (i.e., action) corresponds to negative fitness. This Lagrangian depends upon the flow parameters (and initial states) supplied by the genotype. The agent’s genotype can then be cast as a stochastic gradient descent on negative fitness. This formulation emphasises the relationship between gradients on fitness (selection) and the stochastic terms that are uncorrelated with selection (drift):(12)$$\begin{array}{ccc} \overset{\mathrm{Fast}\;(c.f.,\;\mathrm{phenotypic})\;\mathrm{dynamics}}{\overset{︷}{{\dot{\vec{\alpha }}}_{l}^{\left(i\right)}=D{\vec{\alpha }}_{l}^{\left(i\right)}-\nabla_{{\vec{\alpha }}_{l}^{\left(i\right)}}L_{l}^{\left(i\right)}}} & ⇌ & \overset{\mathrm{Slow}\;(c.f.,\;\mathrm{phylogenetic})\;\mathrm{dynamics}}{\overset{︷}{{\dot{\alpha }}_{n}^{\left(i+1\right)}=(Q_{\alpha_{n}}^{\left(i+1\right)}-\Gamma_{\alpha_{n}}^{\left(i+1\right)})\nabla_{\alpha_{n}^{\left(i+1\right)}}A_{l}^{\left(i\right)}+\omega_{n}^{\left(i+1\right)}}}\\ \underset{\mathrm{Lagrangian}\;(c.f.,\;\mathrm{surprisal})}{\underset{︸}{L_{l}^{\left(i\right)}=L({\vec{\pi }}_{l}^{\left(i\right)}|{\bar{x}}_{l}^{\left(i\right)}\subset \pi^{\left(i+1\right)})}} & & \underset{\mathrm{Action}\;(c.f.,\;\mathrm{adaptive}\;\mathrm{fitness})}{\underset{︸}{A_{l}^{\left(i\right)}=\int dtL{\left(\tau \right)}_{l}^{\left(i\right)}}} \end{array}$$
Formally, the generalised gradient descent at the phenotypic scale corresponds to Bayesian filtering or *inference* [76] that maximises the marginal likelihood of phenotypic paths. This is almost tautological, in that it says that deviations from the most likely developmental trajectory, given some genotype, are unlikely. An additional subtlety here is that the Lagrangian, which plays the role of a Lyapunov function, is a function of sensory states. The implication is that the gradients are not static, but themselves change based upon the way in which the environment interacts with a creature during its development. The stochastic gradient descent at the phylogenetic scale corresponds to Bayesian *learning* via stochastic gradient Langevin dynamics [79], equipped with solenoidal mixing [80].

On this Bayesian reading, phenotypic dynamics *infer* their external dynamics, under a probabilistic model of how external dynamics generate phenotypic dynamics. Intergenerational genetic changes can be seen as *learning* the parameters of a generative model, given the Bayesian model evidence supplied by the scale below (e.g., extended phenotype). This reading rests upon the action (i.e., negative fitness) scoring the accumulated evidence $p(\vec{\pi }|\bar{x})$ for a phenotype’s generative model, $p(\vec{\eta },\vec{\pi }|\bar{x})$ encoded by the extended genotype $\bar{x}$. This evidence is also known as a *marginal likelihood* because it marginalises over external dynamics; i.e., other agents.

**Proof.** The condition in (11) means that the probability of finding an agent of a particular kind is proportional to the likelihood of its phenotypic path, namely, the likelihood a phenotype keeps to the ‘trodden path’, characteristic of the ‘kind’ of ‘things’ that persist. The existence of a nonequilibrium evolutionary steady-state solution to the density dynamics (at both scales) allows us to express the fast and slow dynamics of agents and their autonomous states in terms of Helmholtz–Hodge decompositions. From (1) and (2), we have
(13)$$\begin{array}{ccc} \begin{array}{l} {\dot{\alpha }}_{l}^{\left(i\right)}=(Q_{\alpha }^{\left(i\right)}-\Gamma_{\alpha }^{\left(i\right)})\nabla_{\alpha_{l}^{\left(i\right)}}{ℑ}_{l}^{\left(i\right)}+\omega_{l}^{\left(i\right)}\\ {ℑ}_{l}^{\left(i\right)}=ℑ(\pi_{l}^{\left(i\right)}|{\bar{x}}_{l}^{\left(i\right)}\subset \pi^{\left(i+1\right)}) \end{array} & ⇌ & \begin{array}{l} {\dot{\alpha }}^{\left(i+1\right)}=(Q_{\alpha }^{\left(i+1\right)}-\Gamma_{\alpha }^{\left(i+1\right)})\nabla_{\alpha^{\left(i+1\right)}}{ℑ}^{\left(i+1\right)}+\omega^{\left(i+1\right)}\\ {ℑ}^{\left(i+1\right)}=ℑ(\pi^{\left(i+1\right)}) \end{array} \end{array}$$
The gradients of surprisal at the slow scale, with respect to any given agent’s ‘kind’ or genotype, are the gradients of action by (11):(14)$$\nabla_{\alpha_{n}^{\left(i+1\right)}}{ℑ}^{\left(i+1\right)}=\nabla_{\alpha_{n}^{\left(i+1\right)}}{ℑ}_{n}^{\left(i+1\right)}=\nabla_{\alpha_{n}^{\left(i+1\right)}}A_{l}^{\left(i\right)}$$
Substituting (14) into (13) gives the slow, phylogenetic dynamics in (12) (ignoring certain solenoidal terms). □

For the fast, phenotypic dynamics, we assume that random fluctuations vanish to describe phenotypes that possess classical (i.e., Lagrangian) mechanics, i.e., that are dominated by conservative or solenoidal dynamics. In the limit of small fluctuations, the autonomous paths become the paths of least action, i.e., when the fluctuations take their most likely value of zero. From (4), the autonomous paths of least action are as follows (setting λ=1):(15)$${\dot{\vec{\alpha }}}_{l}^{\left(i\right)}=D{\vec{\alpha }}_{l}^{\left(i\right)}-\nabla_{{\vec{\alpha }}_{l}^{\left(i\right)}}L({\vec{\pi }}_{l}^{\left(i\right)}|{\bar{x}}_{l}^{\left(i\right)})$$
Substituting (15) into (13) gives the fast dynamics in (12).

**Remark** **1.***Note that the extended genotype* ${\bar{x}}_{l}^{\left(i\right)}=\{{\bar{\eta }}_{l}^{\left(i\right)},{\bar{\pi }}_{l}^{\left(i\right)}\}\subset \pi^{\left(i+1\right)}$ *includes the initial states of the extended phenotype. In other words, the extended genotype covers both the genetic and epigenetic specification of developmental trajectories and the initial conditions necessary to realise those trajectories, including external states (e.g., conditions necessary for embryogenesis),*
 $\eta {\left(0\right)}_{l}^{\left(i\right)}\subset {\bar{\eta }}_{l}^{\left(i\right)}$*.*

A useful intuition as to the biological role of the Lagrangian in Equation (11) is that it specifies the states (or trajectories) of a system that has achieved homeostasis. The function will return a small value when physiological measurements are within homeostatic ranges, and increasingly large values as deviations from these ranges become larger. The conditioning upon slow (genotypic) variables means that different sorts of homeostatic ranges are allowable for different sorts of phenotypes. The relationship between the (fast) action and (slow) Lagrangian in Equation (11) implies that phenotypic trajectories—in which homeostasis is maintained—are associated with genotypes that are more likely to be replicated. More precisely, the Lagrangian favours (i.e., its path integral is smaller for) those trajectories in which opportunities for replication are attained—and successful maintenance of homeostasis is only one aspect of this.

The suppression of random phenotypic fluctuations does not preclude itinerant trajectories. Indeed, it foregrounds the loss of detailed balance and accompanying nonequilibria that characterise phenotypic and population dynamics [81,82,83]: for example, biorhythms and chaotic oscillations at the phenotypic scale [84,85,86,87,88] or Red Queen dynamics at the phylogenetic scale [83,89,90]. A system that has the property of detailed balance is one in which time reversal makes no qualitative difference to the dynamics of that system. The implication is that systems in which the solenoidal flow is zero possess detailed balance, while those with a non-zero solenoidal flow do not. The presence of solenoidal flow means that time reversal also leads to a reversal in the direction of this flow. Please see [31] as a relatively recent example of the Helmholtz–Hodge decomposition in Darwinian processes and [80] for a generic treatment of stochastic chaos in this setting. Furthermore, there is no requirement for the grouping operator to return the same partition at each instant of its application. This follows because the grouping operator is determined by sparse coupling among particles at the scale below, which itself may change as certain particles become ‘shielded’ from others [91]: for example, during the self-assembly of particular partitions associated with cell-division, multicellular organisation and development [57]. Mathematically, this permits wandering sets (i.e., partitions) at each scale, where fitness gradients remain well-defined, because they inherit from the dynamics of the scale below.

Implicit in the renormalisation group construction is the notion that variational selection could operate at multiple scales. In other words, although framed in terms of natural selection and evolution, the variational formulation above does not commit to separation of temporal scales apt for replication or reproduction. Any selective mechanism that fulfils the fitness lemma (Lemma 1) will, in principle, be subject to the same selective mechanics. Common examples could include the optimisation of weights in neural networks and their structure learning [45,76,92]. In a biological setting, this selection process could correspond to developmental stages that have well-defined (separation of) temporal scales. Finally, we take a closer look at phenotypic dynamics and explain why they can be construed as sentient behaviour.

## 3. The Sentient Phenotype

An ontological interpretation of phenotypic dynamics—in terms of sentient behaviour or active inference—obtains by expressing the Lagrangian as a *variational free energy*. For clarity, we will drop the sub- and superscripts (and condition on the extended genotype $\bar{x}$) to focus on the generalised states of a given phenotype.

**Lemma** **2.***(Variational free energy): If the autonomous dynamics of a particle or phenotype evince classical (Lagrangian) mechanics, then they can be expressed as minimising a variational free energy functional of Bayesian beliefs—about external states—encoded by their internal phenotypic states,* $p_{\vec{\mu }}(\vec{\eta })$*, under a generative model encoded by their (extended) genotype* $p_{\bar{x}}(\vec{\eta },\vec{\pi }|x_{0})$*:*(16)$$\begin{array}{cc} \dot{\vec{\alpha }} & =D\vec{\alpha }-\nabla_{\vec{\alpha }}F(\vec{\alpha },\vec{s})\\ F(\vec{\alpha },\vec{s}) & =\underset{\mathrm{Energy}\;\mathrm{constraint}}{\underset{︸}{E_{p_{{}_{\vec{\mu }}}}[L_{\bar{x}}(\vec{\eta },\vec{s},\vec{\alpha })]}}-\underset{\mathrm{Entropy}}{\underset{︸}{E_{p_{{}_{\vec{\mu }}}}[L_{\vec{\mu }}(\vec{\eta })]}}\\ & =\underset{\mathrm{Complexity}}{\underset{︸}{D_{KL}[p_{\vec{\mu }}(\vec{\eta })|p_{\bar{x}}(\vec{\eta }|x_{0})]}}+\underset{—\;\mathrm{Accuracy}}{\underset{︸}{E_{p_{\tilde{\mu }}}[L_{\bar{x}}(\vec{s},\vec{\alpha }|\vec{\eta })]}}\\ & =\underset{\mathrm{Divergence}}{\underset{︸}{D_{KL}[p_{\vec{\mu }}(\vec{\eta })||p_{\bar{x}}(\vec{\eta }|\vec{s},\vec{a},x_{0})]}}+\underset{—\;\mathrm{Log}\;\mathrm{evidence}}{\underset{︸}{L_{\bar{x}}(\vec{s},\vec{\alpha })}}\\ L_{\bar{x}}(\vec{x}) & ≜-\ln p_{\bar{x}}(\vec{x}|x_{0}) \end{array}$$

This variational free energy can be rearranged in several ways. First, it can be expressed as an energy constraint minus the entropy of the variational density, which licences the name free energy [93]. In this decomposition, minimising variational free energy corresponds to the maximum entropy principle, under the constraint that the expected Lagrangian is minimised [51,94]. The energy constraint is a functional of the marginal density over external and sensory states that plays the role of a generative model (i.e., parameterised by the extended genotype), namely, a joint density over causes (external dynamics) and their consequences (autonomous dynamics). Second—on a statistical reading—variational free energy can be decomposed into the (negative) log likelihood of particular paths (i.e., *accuracy*) and the KL divergence between posterior and prior densities over external paths (i.e., *complexity*). Finally, it can be written as the negative *log evidence* plus the KL divergence between the variational and conditional (i.e., posterior) density. In variational Bayesian inference [95], negative free energy is called an *evidence lower bound* or ELBO [96,97,98].

**Proof.** The sparse coupling—that underwrites a particular partition—means autonomous paths (i.e., generalised states) depend only on sensory paths. This means there is a (deterministic and injective) map from the most likely autonomous paths (of sufficiently high order generalised motion) to the conditional density over external paths, where both are conditioned on sensory paths. This injection means we can consider the conditional density over external paths as being parameterised by internal paths. We will call this a *variational density* (noting from (6) that internal paths are conditionally independent of external paths):
(17)$$\begin{array}{cc} p_{\vec{\mu }}(\vec{\eta }) & ≜p_{\bar{x}}(\vec{\eta }|\vec{s},\vec{a},x_{0})=p_{\bar{x}}(\vec{\eta }|\vec{s},\vec{a},\vec{\mu },x_{0})\\ \vec{\alpha } & =\arg\min_{\vec{\alpha }}{ℒ}_{\bar{x}}(\vec{\alpha }|\vec{s})\Rightarrow \\ \vec{\mu } & =\arg\min_{\vec{\mu }}{ℒ}_{\bar{x}}(\vec{\mu }|\vec{s},\vec{a})\\ \vec{a} & =\arg\min_{\vec{a}}{ℒ}_{\bar{x}}(\vec{a}|\vec{s},\vec{\mu }) \end{array}$$
This definition means that the Lagrangian and variational free energy share the same minima, where their gradients vanish:(18)$$\begin{array}{l} \vec{\alpha }=\arg\min_{\vec{\alpha }}L_{\bar{x}}(\vec{\alpha }|\vec{s})=\arg\min_{\vec{\alpha }}F(\vec{\alpha },\vec{s})\\ \;\Rightarrow \nabla_{\vec{\alpha }}L(\vec{\alpha },\vec{s})=0\\ \;\Rightarrow \nabla_{\vec{\alpha }}F(\vec{\alpha },\vec{s})=\nabla_{\vec{\alpha }}\underset{\mathrm{Divergence}=0}{\underset{︸}{D_{KL}[p_{\vec{\mu }}(\vec{\eta })||p_{\bar{x}}(\vec{\eta }|\vec{s},\vec{a})]}}+\underset{=0}{\underset{︸}{\nabla_{\vec{\alpha }}L_{\bar{x}}(\vec{s},\vec{\alpha })}}=0 \end{array}$$
If autonomous dynamics are conservative, their trajectory is a path of least action and we can replace the Lagrangian gradients in (12) with variational free energy gradients to give (16). □

**Remark** **2.**
*The free energy lemma (Lemma 2) associates negative fitness with variational free energy, such that phenotypic behaviour will appear to pursue paths of least free energy or greatest fitness. Because variational free energy is an upper bound on log evidence, the pursuit of maximum fitness can be read as self-evidencing [99]: namely, actively soliciting evidence for generative models endowed by evolution. In short, autonomous dynamics (appear to) actively infer external states under a generative model, whose parameters are (apparently) learned by minimising a path integral of variational free energy.*


The functional form of variational free energy licences a teleological interpretation of autonomous dynamics; the internal paths can be read as the sufficient statistics or parameters of (approximate) Bayesian beliefs about external states, while active paths will (appear to) change the posterior over external states to ‘fit’ internal (Bayesian) beliefs. In other words, active dynamics will look as if they are trying to fulfil the predictions of internal representations. A complementary interpretation inherits from the decomposition of variational free energy into complexity and accuracy. Minimising complexity means that generalised internal states encode Bayesian beliefs about external states that are as close as possible to prior beliefs, while generalised active states will look as if they are changing sensory states to realise those beliefs. These interpretations—in terms of *perception* and *action*—furnish an elementary but fairly expressive formulation of active inference. For example, the free energy formulations above have been used to emulate many kinds of sentient behaviour, ranging from morphogenesis [100], through action observation [101], to birdsong [102].

Although not developed here, the renormalisation group construction means that we can apply the same arguments to autonomous kinds—i.e., agents—at the slow scale. In other words, on average, the extended genotype of internal kinds comes to encode Bayesian beliefs about external kinds, while active kinds will look as if they are trying to realise those beliefs, via niche construction [77,103,104,105]. In virtue of the minimisation of variational free energy, we have an implicit maximum entropy principle, which brings us back to [21,22] via [49].

## 4. Variational Recipes

Effectively, we are describing the evolutionary developmental process with the following protocol:
First, generate an ensemble of particles (i.e., extended phenotypes) by sampling their flow parameters and initial states (i.e., extended genotypes) from some initial density.For each particle, find the path of least action using a generalised Bayesian filter (i.e., active inference).After a suitable period of time, evaluate the path integral of variational free energy (i.e., action) to supply a fitness functional.Update the flow parameters and initial states, using a stochastic gradient descent on the action (i.e., Darwinian evolution).

If this protocol were repeated for a sufficiently long period of time, it would converge to an attracting set, assuming this pullback attractor exists [32]. In statistical mechanics, this would be a nonequilibrium steady state, while in theoretical biology, it would correspond to an evolutionary steady state, at a certain timescale.

The notion of a steady state is clearly an idealization, as it assumes an unchanging environment. The local environments of all organisms are, however, moving targets, largely due to the activities of other organisms. Even if all of Life is considered a single population, it faces a changing local (i.e., biospheric) environment due to its—Life’s—own activities, as well as to bolide impacts and other abiotic causes. Hence, we can expect evolution to remain always ‘in process’ even for large, diverse populations. The assumption of an asymptotic evolutionary steady state is, therefore, effectively an assumption of a local (in time) steady state that has a lifetime long enough for evolutionary processes to be significant but short enough that the local environment of the evolving system can be considered approximately fixed. We now conclude with a simple application of the above protocol to a special case of selection in neurobiology.

### A Numerical Study of Synaptic Selection

Figure 3 shows the results of a numerical study of selection processes, using the variational procedures above. This example illustrates the interplay between minimising variational free energy over somatic lifetimes and its use in selecting phenotypes at a slow, transgenerational, timescale. This example considers a relatively straightforward selection process in neurobiology, namely, synaptic selection in neurobiology, which illustrates the nested scales over which free energy minimising processes evolve. Specifically, we simulated a single neuron (i.e., nerve cell) immersed in an environment constituted by potential pre-synaptic inputs in the surrounding neuropil. Unbeknown to the neuron (or more specifically, its dendritic tree), these presynaptic inputs fluctuated systematically with spatially structured waves of activation. These waves could only be detected by deploying postsynaptic specialisations (i.e., sensory states) in an ordered sequence along the dendrite. The details of this simulation are not important, and can be found in [106]. The key point here is that the cell’s adaptive fitness—read as negative variational free energy—depends upon predicting its synaptic inputs through internal, intracellular dynamics that recapitulate the external, extracellular or environmental generation of sensory (synaptic) inputs. However, to do this, the dendrite has to have the right morphology, parameterised by the location of synapses on the dendritic surface.

To model learning and inference, the synapses were rendered more or less sensitive to their presynaptic inputs by optimising their sensitivity (a.k.a., precision) with respect to variational free energy in a biologically plausible fashion (i.e., using electrochemical equations of motion that performed a gradient flow on variational free energy). This meant that as the cell accumulated evidence from its presynaptic environment, its free energy decreased, and it became better at predicting its presynaptic inputs. However, this ability to predict depends upon selecting synapses that are located in the right order, along the dendrite.

To simulate synaptic selection, we used Bayesian model selection to compare the evidence for a cell’s model with and without a particular synaptic connection. If the free energy increased, the postsynaptic specialisation was moved to another location at random. This process was repeated to simulate slow (Bayesian model) synaptic selection, until the phenotypic morphology of the dendrite was apt for accurately modelling (i.e., fitting) the waves of pre-synaptic input. In this example, the Bayesian model selection used Bayesian model reduction [107], based upon the optimised sensitivity (i.e., precision) of each synapse: very much along the lines of synaptic regression and implicit homeostasis [108,109,110]. Figure 3 shows the progressive reduction in free energy at a slow timescale as the synapses that enable the cell to predict or fit its environment are selected.

**Figure 3 entropy-25-00964-f003:**
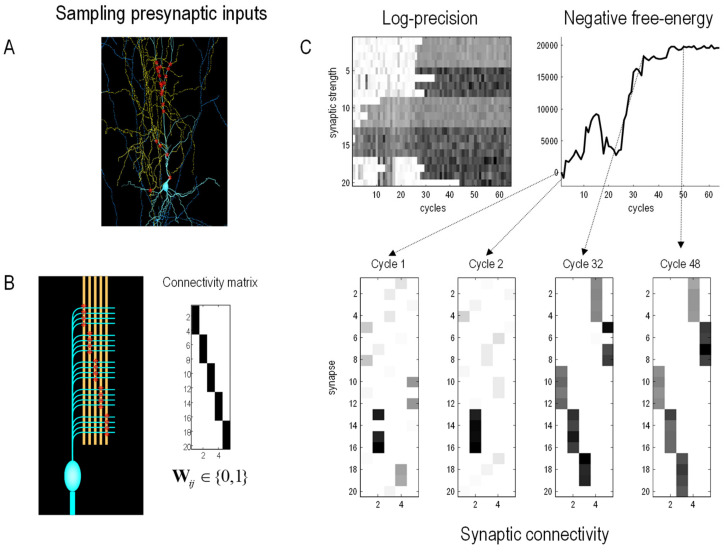
Synaptic selection. This figure reports the results of numerical studies using fast free-energy minimising processes to model phenotypic dynamics and slow free-energy minimising processes to select phenotypic configurations or morphologies that, implicitly, have the greatest adaptive fitness or adapt to fit their environment. In this example, we focus on the selection of synapses of a brain cell (i.e., neuron) that samples presynaptic inputs from its neuropil (i.e., environment). The details of the generative model—used to simulate intracellular dynamics as a gradient flow on variational free energy—can be found in [107]. The key thing about these simulations is that—after a period of time—certain synapses were eliminated if Bayesian model selection suggested that their removal increased Bayesian model evidence (i.e., decreased variational free energy). (**A**): Findings in [111] suggest that neurons are sensitive to the pattern of synaptic input patterns. The image shows a pyramidal cell (blue) sampling potential presynaptic inputs from other cells (yellow) with postsynaptic specialisations (red). (**B**): In this model, pools of presynaptic neurons fire at specific times, thereby establishing a hidden sequence of inputs. The dendritic branch of the postsynaptic neuron comprises a series of segments, where each segment contains a number of synapses (here: five segments with four synapses each). Each of the 20 synapses connects to an axon of a specific presynaptic pool. These provide presynaptic (sensory) inputs at specific times over the length of a dendrite. If each of the 20 synapses were deployed in an orderly fashion across the five segments—as in the connectivity matrix—an orderly sequence of postsynaptic activations would be detected, and, implicitly. (**C**): The lower panels show the deployment of synaptic connections over 64 ‘generations’ (i.e., cycles), in which the precision (a.k.a. sensitivity) of synapses was used to eliminate synapses if they did not contribute to model evidence. Each ‘lifetime’ of the cell was 120 (arbitrary) time units, during which time two waves of activation were detectable. The upper panels score the ensuing increase in marginal likelihood or adaptive fitness (negative free energy) over the 64 generations. The left panel shows the accompanying increase in the sensitivity (i.e., log-precision) of the 20 synapses as they find the collective arrangement that maximises adaptive fit or model evidence for this (neuronal) environment.

## 5. Discussion

One insight from the above analysis is that populations are not necessarily quotient sets of equivalence classes. Put simply, there is no assumption that any given particle shares phenotypic or genotypic characteristics with any other particle. This observation is interesting on two counts. First, it suggests that treating a population as an equivalence class of conspecifics may not be sufficient, in the sense that the population includes all of the (natural) kinds that interact to maintain their particular partition. The fact that all ‘individual’ multicellular eukaryotes appear to be holobionts—effectively, complex, multispecies ecosystems—bears this out [95,96]. The ‘genotype’ of such a system is a probability distribution of probability distributions, each of the latter over one of the component ‘species’ composing the holobiont. The phenotype of the holobiont, including its reproductive success and hence ‘fitness’ in the narrow reading, is a function of this bilevel probability distribution. Differential rates of genetic change between component genomes—and the fact that actions at the phenotypic level can alter the genotype as a probability distribution (e.g., humans can take anti- or probiotics)—complicate the difference in characteristic times assumed in Lemma 1, as discussed further below. Second, even if some agents share the same genotype, their phenotypes can specialise in distinct ways to minimise their joint variational free energies. This is obvious in the case of multicellular eukaryotes, all of which exhibit differentiation of cellular phenotypes during morphogenesis; see [89] for a worked example specifically employing the FEP formalism, and [97] for simulations demonstrating that multicellularity with differentiation provides a generic means of minimising VFE from the environment. These considerations together mandate a quintessentially co-evolutionary perspective that emphasises co-dependencies and co-creation [16,98,99,100].

However, the emergence of equivalence classes—e.g., ‘species’ of holobionts—begs explanation. A potential answer is the generalised synchrony between particles, as they find their joint variational free energy minima—and become mutually predictable; e.g., [52,91]. In an evolutionary setting, one can imagine the search for joint variational free energy minima leading to convergent evolution or speciation (Luc Ciompi, personal communication; [101]). Reproduction is, in all extant organisms, a matter of cell division, and closely related cells reap a free-energy advantage by working together [97]. An effective—though metabolically, morphologically, and behaviourally expensive—mechanism to protect this advantage is sex. The proliferation of species-specific morphological and behavioural specializations, together with the suppression of stem-cell pluripotency required to render sex obligate [102] in ‘higher’ eukaryotes, attests to the success of this strategy. From the present perspective, sex is a particularly elaborate feedback pathway—from the phenotypic to the genotypic scale—that preserves the integrity of the latter. It is, in other words, a mechanism that decreases VFE for the genome at the expense of increased VFE for the phenotype.

The synthesis of biological evolution and development on offer here is an example of a generalised synthesis: applicable, under the free energy principle, to all kinds of things. This synthesis can be read as generative models autopoietically generating entities and then using the ‘fit’ of the model to the niche as evidence for updating the model, in a cyclical process summarised in Figure 4.

## 6. Limitations

As with most applications of the free energy principle, the variational account alone does not supply a process theory. Rather, it starts from the assumption that a nonequilibrium (evolutionary) steady state exists and then describes the dynamics that the system must exhibit. Thus, the variational account enables various process theories to be proposed as specific hypotheses about biological systems. For example, the genetic variation in the above formulation follows from the Helmholtz decomposition or fundamental theorem of vector calculus. However, the ensuing stochastic gradient Langevin dynamics does not specify the particular processes that give rise to this kind of dynamics, e.g., [103]. There are many candidates one could consider: for example, simple rejection sampling or more involved genetic algorithms that provide a plausible account of bisexual reproduction [104,105]. A computationally expedient way of evaluating the requisite gradients—for example those for simulating artificial evolution—could call upon Bayesian model reduction [45,112]. Irrespective of the replication or reproduction process, it must, on the present analysis, conform to a stochastic gradient flow on ‘fitness’ with solenoidal mixing [72,78,79].

This openness to multiple process theories is an advantage of the current approach, both in convergence situations in which diverse genomes produce very similar phenotypes [112] and in the complementary situations in which a single genome supports diverse phenotypes. Neither situation is rare: genomes as different as those of *Amoeba proteus* and *Homo sapiens* can produce amoeboid cells, and the differentiated cells of any multicellular organism illustrate phenotypic diversity at the cellular level. While the general theory outlined here merely requires that some process exists, we can realistically expect one-to-many process mapping in both directions when dealing with real biological systems.

The primary offering of this variational formulation of natural selection—from an empirical perspective—is that one can hypothesise alternative forms for the Lagrangian. Each choice of Lagrangian will have consequences not only for the dynamics over physiological and developmental timescales but will also allow for predictions as to evolution over phylogenetic timescales. It is also worth noting that the account of natural selection set out here, in which genotypic evolution depends upon the action of phenotypic paths, applies to systems that satisfy the variational fitness lemma (Lemma 1): namely, the likelihood of an agent’s genotype corresponds to the likelihood of its phenotypic trajectory. While a plausible assumption—that is intuitively consistent with Darwinian evolution—we did not examine the conditions under which this assumption holds. This means there is an opportunity to further the ideas set out in this paper by examining the sorts of stochastic systems in which the variational fitness lemma (Lemma 1) holds. It could be argued that Lemma 1 must hold at least in those systems where the genotype transforms into the phenotype retaining an equivalence within stochastic limits. For example, gene expression is the most fundamental level at which the genotype gives rise to the phenotype, and this mapping from genotype to phenotype is the subject of the many process theories studied by developmental biology. On a teleological view, one might further argue that active inference is necessary to maintain a high degree of equivalence during the course of this transformation and to preserve a correspondence between genotype and phenotype.

Available edge cases are, however, informative. Single mutations can induce saltatory changes in phenotype; a canonical example is the four-winged *Drosophila melanogaster* fly produced by combining three mutations, *abx, bx^3^*, and *pbx* of the bithorax complex in a single animal [113]. In complementary fashion, the planarian *Dugesia japonica* reproduces by fission followed by regeneration and has a heterogeneous, mixoploid genome with no known heritable mutants [114]; the phenotype of this animal has, however, remained stable for many thousands of generations in laboratories, and in all likelihood for millions of years in the wild. The phenotype can, moreover, be perturbed in saltatory fashion from one-headed to two-headed by an externally imposed bioelectric change; this altered phenotype is bioelectrically reversible but otherwise apparently permanent [115]. Engineering methods can create even more radically diverse phenotypes without genetic modifications, as demonstrated by the ‘xenobots’ prepared from *Xenopus laevis* skin cells, which adopt morphologies and behaviours completely unlike those that skin cells manifest when in the frog [116,117].

The availability of experimentally tractable edge cases of Lemma 1 provides an opportunity to further the ideas set out in this paper by examining the sorts of stochastic systems in which the variational fitness lemma (Lemma 1) holds. The kinds of edge cases mentioned above suggest, however, that Lemma 1 could be weakened to holding ‘up to’ saltatory events, including abiotic events such as bolide impacts, affecting genotype, phenotype, or both without substantially affecting the theory. Any systems that survive such events—any systems whose Markov blankets remain intact—simply carry on, undergoing learning, variation, and selection as if the saltatory event had never occurred.

One could suggest that Lemma 1, and the broader scope of the formalisms described here, may be applicable to systems where a population of entities engages in intergenerational replication (modelled here using the renormalisation operations), and where those entities at a faster timescale engage in rapid adaptation (e.g., development, learning, behaviour, modelled with active inference) during their lifetime. These two levels could, for example, model how genome-based intergenerational evolution sets initial conditions for organismal molecular and behavioural developments. For the faster intra-generational scale, the external states model the material basis of what the phenotype is a generative model of. For the slower inter-generational scale, the external states are updated through time as a process of renormalisation (reduction and grouping) of the extended genotype-phenotype.

## 7. Conclusions

This work attempts to unify the slow, multi-generational phylogenetic process of natural selection with the single-lifetime, phenotypic process of development (equations and notation summarized in Appendix A). In this perspective, a bidirectional flow of information occurs as evolution imposes top-down constraints on phenotypic processes, and action selection provides evidence that is selected for by the environment (i.e., bottom-up causation). In this account, learning and inference occur through updating probabilistic beliefs via Bayesian model selection in evolutionary time and active inference in developmental time. The fitness of (extended) genotypes and (extended) phenotypes is selected for through the minimisation of the same free energy functional: Bayesian model evidence or marginal likelihood.

Further studies using both simulations and laboratory experiments are clearly needed to test this framework in the context of particular process theories that propose explicit functional connections between genotype and phenotype. While Lemma 1 is *prima facie* plausible in the case of idealised ‘central dogma’ organisms in which phenotype is largely determined by genotype within a tightly constrained, essentially static niche, the relation between genotype and phenotype in holobionts inhabiting realistic niches can be expected to be substantially more complex. ‘Egalitarian’ organisms, e.g., obligate symbionts or holobionts, comprising cells with different genotypes [118] and engineered systems—that offer cells radically different environments than they have experienced in phylogenetic evolution to date [119]—may be of particular interest for such studies.

## Figures and Tables

**Figure 1 entropy-25-00964-f001:**
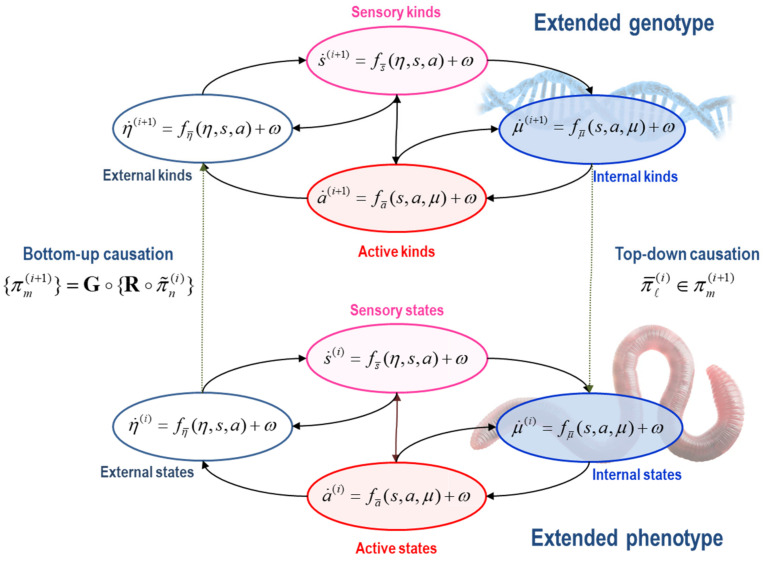
Schematic (i.e., influence diagram) illustrating the sparse coupling among states that constitute a particular partition at two scales.

**Figure 2 entropy-25-00964-f002:**
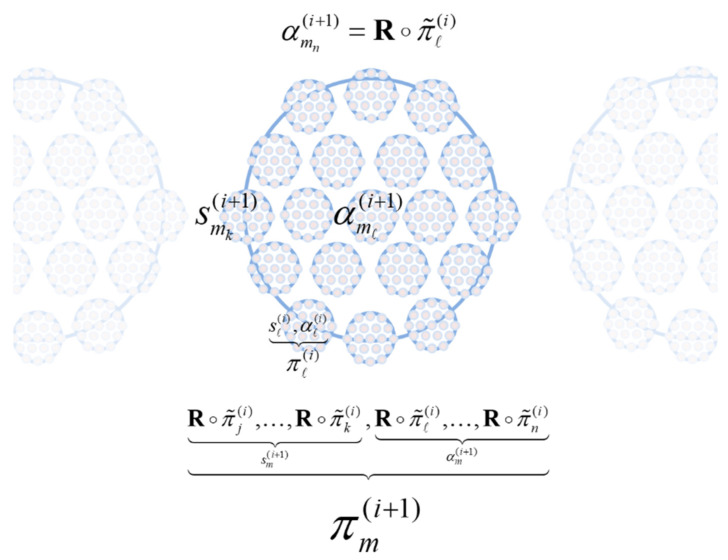
Schematic showing the hierarchical relationship between particles at scales *i* and *i* + 1. For clarity, sensory and autonomous states are illustrated in blue and pink, respectively. Note that each variable is a (very large) vector state that itself is partitioned into multiple vector states. At scale *i* + 1, each particle represents an ensemble (e.g., $\pi_{m}^{\left(i+1\right)}$ is population *m*), the elements of which are partitioned into autonomous and sensory subsets (e.g., $\alpha_{m_{n}}^{\left(i+1\right)}$ is the *n*-th autonomous genotype from population *m*). At scale *i*, each particle represents an element of an ensemble (e.g., $\pi_{l}^{\left(i\right)}$ is the l-th phenotype), which is itself partitioned into sensory and autonomous subsets. The slow states of each element (e.g., phenotype) are recovered by the reduction operator **R**, to furnish the states at the ensemble level (e.g., genotype). A key feature of this construction is that it applies recursively over scales.

**Figure 4 entropy-25-00964-f004:**
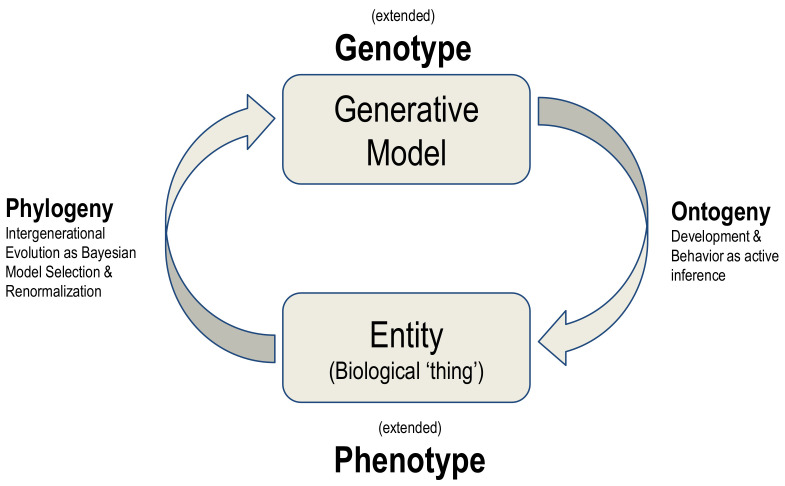
Phylogeny and ontogeny as bottom-up and top-down causation.

## Data Availability

All data are included in the manuscript and Appendix A.

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
