# Peer review of "A Variational Synthesis of Evolutionary and Developmental Dynamics"

_entropy, 2023, doi:10.3390/e25070964_

Round 1

Reviewer 1 Report

The authors present a very general theory based on stochastic processes to describe natural selection. The subject is interesting and the effort to include all the relevant underlining mechanisms at play in a unified framework is commendable. However, the authors do not present any explicit example to illustrate their theory with concrete applications. The whole discussion is too general and therefore remains too vague. As far as I have understood, for instance by looking at Figure 1, the theory features a set of coupled Langevin equations describing the evolution of the relevant variables in the system.  Then some formal results are presented, but without examples of applications to real systems, or more specific models, it is really hard to follow the authors' discussion. Therefore, I do not recommend publication.   

Author Response

We have gone through the manuscript and packing difficult to read sentences. Furthermore, we have substantially extended the discussion, relating the technical ideas to examples in theoretical biology (please see below). Finally, we have created a tabular glossary detailing the technical terms and mathematical expressions that will hopefully make it easier for readers to follow the arguments.

Please see the attachment for Reviewer 4, where all comments are addressed. 

Reviewer 2 Report

A variational method to natural selection has been presented by the authors. The ideas presented in the article (which is mostly mathematical) present a way to formalize adaptive fitness as a function of phenotypic fitness and given the novelty of approach, should be published.

The only comment I have is the authors should consider some numerical analysis and simulations to support their ideas.

Author Response

Thank you for these suggestions. We have now noted the importance of numerical analyses and simulations in the concluding paragraph (please see below). Although some of the theoretical procedures have already been applied piecemeal in the literature, an end to end simulation of these ideas would be a substantial undertaking that would overwhelm the current paper. We have therefore elected to illustrate the ideas using biological examples in the discussion, in the hope that this provides support for the technical ideas. (Please see the additions at the end of this remittal letter).

Please see the attachment for Reviewer 4, where all comments are addressed. 

Reviewer 3 Report

Attached pdf

Author Response

Thank you for these comments.

Comments on the manuscript format.

Although the authors do an excellent job on the technical part of the manuscript and follow the rule that the word "This" should be followed by a noun, there one too many instances where the authors did not follow "This" with a noun, which the reading awkward and confusing.

Thank you. We have now gone through revising these improper sentences. Please see the list of revisions below.

Why is that the authors use bullets? The authors should use other labels but not bullets for a list of concepts.

No problem. We have now removed the bullets.

Through the ms, the authors use the future tense instead of the present tense. In particular, the future tense is awkward because there are no examples with quantitative results at the end of the ms. Although the technical aspects developed by the authors publishable, it is unclear why they do not include at least one example and discuss the quantitative results. Do encourage the authors to include in the revision at least one practical example where the reader can follow the details of the analysis.

We have now noted the importance of numerical analyses and simulations in the concluding paragraph (please see below). Although some of the theoretical procedures have already been applied piecemeal in the literature, an end to end simulation of these ideas would be a substantial undertaking that would overwhelm the current paper. We have therefore elected to illustrate the ideas using biological examples in the discussion, in the hope that this provides practical examples — at least of an intuitive kind. (Please see the additions at the end of this remittal letter).

Please see the attachment for Reviewer 4, where all comments are addressed. 

Reviewer 4 Report

Authors provide in the paper mathematical formulation of the natural selection with use of variational analysis.  It is in scope of the Entropy journal.

Introduction is well written and explains the goal of the work. In the next section, variational formulation is described.
Here, to improve explanations of used symbols, please provide after every equation lists of symgols used with explanation, for the reader convenience.

In the last subsection natural selection variational formalism is introduced. Next section provides variational free energy formalism for the phenotype dynamics.  Paper contain section with discussion, pointing here limitations of postulated models.

The weak point of the paper, already mentioned by authors in the abstract is lack of any calculated results obtained basing on provided formalism. However, despite this paper can be published, as the first in the series - where following ones should include results which support usefullness of postulated models.

Author Response

Thank you for these comments. We have now provided supplementary material in the form of a tabular glossary of technical terms of mathematical expressions (please see the link at the end of this remittal letter). We hope that this will help readers navigate the equations.

Furthermore, we have now noted the importance of numerical analyses and simulations in the concluding paragraph (please see below). Although some of the theoretical procedures have already been applied piecemeal in the literature, an end to end simulation of these ideas would be a substantial undertaking that would overwhelm the current paper. We have therefore elected to illustrate the ideas using biological examples in the discussion, in the hope that this illustrates the potential usefulness of the variational formulation. (Please see the additions at the end of this remittal letter).

Round 2

Reviewer 1 Report

The authors did not address the main comment from me and the other Referees, namely the absence of any explicit example with numerical or analytical computations, or applications to real data, to check and illustrate the proposal framework. They only added a final paragraph where the need for such a further study is evidenced.

Therefore, I do not recommend publication of the paper.  

Author Response

Thank you, addressed in the remittal letter. 

Reviewer 4 Report

Changes make text easier to follow by the reader. Before publication, please check supplementary material equations sizes, in the present form it is not well done by technical editing standards.

I am supporting publishing of the paper in the current shape. However, I strongly recommend to soon publish results basing on the described methodology.

Author Response

Thank you, see remittal letter. 
